# Is Preoperative Asymmetry a Predictor of Postoperative Asymmetry in Patients Undergoing Breast Reduction?

**DOI:** 10.3390/ijerph20053780

**Published:** 2023-02-21

**Authors:** Tomasz Zawadzki, Bogusław Antoszewski, Anna Kasielska-Trojan

**Affiliations:** Plastic, Reconstructive and Aesthetic Surgery Clinic, Medical University of Lodz, Kopcinskiego 22, 90-153 Lodz, Poland

**Keywords:** breast reduction, asymmetry, volume

## Abstract

Plastic surgeons aim to achieve symmetry in breast surgeries, which is the main determinant of chest aesthetics. The aim of this study was to verify if preoperative breast asymmetry is a predictor of postoperative asymmetry in women undergoing breast reduction. In this prospective study, we enrolled 71 women (the mean age 37 years, SD 10 years) with breast hypertrophy who underwent reduction mammaplasty. We collected clinical data including age, height, weight, weight of the resected tissues, and performed pre and postoperative photographic documentation. The following measurements of both breasts were analyzed: volumes (vol), nipple–sternal notch distance (A-sn), difference between nipples’ levels (A-A’), nipple–midline distance (A-ml), difference between inframammary folds levels (IF-IF’), distance between inframammary fold and nipple (IF-A), distance between inframammary fold apex and midline (IF-ml). All measurements were performed preoperatively and 6 months after the surgery and asymmetries of all variables were calculated (asy-vol, A-A’, asyA-sn, asyA-ml, IF-IF’, asyIF-A, asyIF-ml). Postoperative asymmetry of breast volumes and nipples position was not associated with any of the analyzed clinical variables. Postoperative asymmetry of nipples’ level was associated with preoperative asymmetry of IF-ml; however, logistic regression did not detect any preoperative measurement influencing postoperative volume and nipples’ level asymmetry. Moreover, we found that preoperative asyIF-ml increased the risk of postoperative volume asymmetry, which is above the average (52 cc) (OR = 2.04). Postoperative breast asymmetry after breast reduction is not related neither to preoperative asymmetries nor clinical variables; however, asymmetry of inframammary fold apex to the midline may be the factor affecting postoperative volumetric asymmetry.

## 1. Introduction

Breast hypertrophy is a disabling condition characterized by excessive breast tissue growth. There is no universal definition and/or classification of gigantomastia. Some authors used to define it as breast enlargement that requires reduction of over 1500 g per breast. There is, however, discordance in the literature regarding criteria including the weight of resected tissued ranging from 800 g to 2000 g; there were also some suggestions to consider a certain bra cup size as a criterium of diagnosis. However, such extend resections appeared to be quite rare and limited the possibility of diagnosing the condition. Nowadays, the major diagnostic criteria for gigantomastia are not the certain volumes of excessive breast tissues which need to be removed but subjective psychological and mostly physical symptoms such as: mastalgia, back pain, postural problems, chronic traction injury to 4th/5th/6th intercostal nerves that can cause loss of nipple sensation, and inframammary fold dermatoses that can cause infections and even ulcerations. In addition, hypertrophic breasts are often more asymmetrical than normal-sized breasts, which can further exacerbate these symptoms. The etiology of gigantomastia in most cases is unknown (idiopathic), but it can be the result of hormonal disturbances and changes due to some disorders, or it may arise from physiological conditions such as pregnancy (gestational gigantomastia) [1,2].

According to ISAPS statistics from 2021, reduction mammaplasty is in 10th place of worldwide surgical procedures performed by plastic surgeons, with a 19% increase compared to 2020. This shows the growing demand for these procedures among women worldwide and forces surgeons to strive for the improvement of their outcomes. 

Plastic surgeons aim to achieve symmetry in all breast surgery procedures, which is the main determinant of chest aesthetics. In the past 100 years since the first breast reduction technique was described by Max Thorek [3] (which involves transposition of nipple—areola complexes (NAC) as free skin grafts), many new surgical methods have been reported. Although this technique is simple, reliable, and still applicable in a selected group of patients, according to the literature the main disadvantage of Thorek’s method is the tendency for wide and flat breasts, unpredictable nipple projection, and poor pigmentation of the areolas (hypopigmentation and discoloration). Historically, an important step forward for modern breast reduction surgery was a new skin resection pattern designed by Robert Wise in 1956, which became known as the inverted T or keyhole pattern, which enabled detailed measurement-based preoperative planning [4]. Over the years, significant advances in pedicle techniques were achieved by many surgeons: Pitanguy in the 1960s, who introduced the superior pedicle; his technique was further refined by Orlando and Guthrie in 1975 with the incorporation of more medial parenchyma into the pedicle to ensure adequate vascularity of the nipple–areolar complex; McKissock in 1972, with his design of a vertical bipedicle; Robbins in 1977; and Courtiss and Goldwyn in 1977, who described the inferior pedicle. These pedicles were adapted to the Wise skin resection pattern and have been widely used by plastic surgeons all around the world [5,6]. Although the inverted-T technique is very popular, it has several significant disadvantages: “boxy” appearances of breasts, the scars are extensive especially along the inframammary folds, and, over time, breast shape changes start to occur such as loss of upper pole fullness and pseudoptosis (bottoming out). For this reason, some surgeons are still looking for further modifications of these techniques to overcome their limitations, especially in cases of “smaller” breast reductions. However, they still claim the inverted-T pattern as irreplaceable for larger breast reductions [7]. A focus on eliminating the horizontal scars’ lengths while using the superior pedicle was popularized by Lassus and further modified and described by Lejour in 1970 [8,9]. The vertical skin incision pattern (without inframammary horizontal incision) has been adapted to inferior, medial, superomedial, and lateral pedicles [5]. Lejour’s technique with a superior pedicle and vertical scar closure has been widely used and many women have obtained very good results despite few disadvantages such as: an ill-defined inframammary fold and poor nipple–areola complex sensitivity. The recent modern approach for breast reduction involves preserving anatomical NAC innervation and vascularity. Description (by Würinger in 1998) of the thin fibrosus septum which contains nerves and vessels supplying the nipple–areola complex enabled another step forward in breast reduction surgeries (septum-based breast reductions) [10,11,12]. All these methods have been developed to improve the variables which add to the “perfect breast” such as: breast shape, footprint, and nipple–areolar complex (NAC) sensation and projection for both better aesthetic and functional results. [4,5,6,7,8,9,11,12,13,14]. Furthermore, it seems to be important to verify the available surgical techniques in the aspect of the possibility to achieve “clinically” symmetric breasts.

The dimensions and proportions of “aesthetically perfect” breasts are well known, and they are determined based on volume, shape, nipple position, projection, relation to the other breast, and patients’ bodies. [6,15,16,17,18,19]. BREAST-Q form is currently the most frequently used survey to verify the impact of breast surgery on the patient’s life quality and her satisfaction from the procedure, from the patient’s perspective. In one of the domains evaluating patient satisfaction, apart from questions about the appearance of the breasts, there is a question about self-assessment of the symmetry of breasts [20]. Therefore, obtaining postoperative symmetry is one of the key goals determining the patient’s satisfaction regarding the aesthetic effect of the breast procedure.

Detailed preoperative planning that relies on direct measurements of breasts is performed to achieve postoperative “clinical” breast symmetry which can be verified by many techniques such as direct and indirect anthropometry, 3D scanning, CT, MRI, mammography, and water displacement techniques [21,22,23]. All these techniques have many advantages and disadvantages. The most common disadvantages are: limited accessibility for the doctor and patient, high costs of equipment, the need for a trained personnel for their exploitation and interpretation of results, and the time needed to obtain the desired metric data. Due to the lack of widely available and easy-to-use tools, the Breast Idea (BI) application was created. It enables quick assessment of breasts (their measurements and asymmetries) based on photographic documentation, which may be helpful in a breast surgeon’s clinical practice [24,25]. These techniques can be also helpful not only to evaluate postoperative outcomes, but also as an addition for more complex preoperative planning. They can detect and measure primary asymmetry of different variables allowing surgeons to better plan breast symmetrization procedures, choose the right implants, often of different volumes, and project for aesthetic augmentation or breast reconstruction after oncological treatment. Furthermore, asymmetrical resections of tissues in reduction mammaplasty procedures may be considered to achieve postoperative clinical symmetry.

Breast reduction is one of the most complex breast procedures and carries the risk of minor or major complications. For patients undergoing different types of breast reduction procedures, there are lots of factors increasing the risk of postoperative complications, which are reported in 7–20%, such as: wound dehiscence, NAC necrosis, seroma, hematoma, and excessive scarring. The most common factor, that increase the mentioned complications, are excessive weight of the resected tissues and BMI > 30 kg/m^2^ [26,27,28,29]. There are no studies analyzing the influence of these factors on postoperative breast asymmetry, and this aspect is not often mentioned in published articles of breast reduction outcomes, although management of postoperative breast asymmetry is challenging [30].

The aim of this study was to verify if preoperative breast asymmetry is a predictor of postoperative asymmetry in women undergoing breast reduction. We focused on analyzing clinical variables’ influence and preoperative breast measurement asymmetries’ effect on the postoperative asymmetry of breast volume and nipples positions.

## 2. Materials and Methods

In this prospective study, 71 patients who underwent bilateral reduction mammaplasty between 2019 and 2022 in the Plastic, Reconstructive, and Aesthetic Surgery Clinic and met the study criteria were included. We collected clinical data including age, height, weight, weight of the resected tissues, and performed pre and postoperative (6 months after the surgery) photographic documentation. Inclusion criteria were: a patient qualified for the Wise pattern bilateral reduction mammaplasty with superomedial dermal pedicle (random vascularization) for the transposition of the NAC with the gland resection caudal to the Wuringer’s septum in the inferior part of the breast followed by a triangular resection lateral to the pedicle, surgery performed by the same team of surgeons under general anesthesia, patient’s BMI < 30 kg/m^2^, available for the follow-up visit 6 months after the procedure, and no major postoperative complications (nipple–areola complex necrosis, breast hematoma).

### 2.1. Measurements 

All measurements were performed based on the patients’ photographs performed according to the instructions of the web application Breast Idea (BI) (Figure 1a,b and Figure 2a,b) Two modules of the application (Quick Assessment and Volume Estimator—BIVE) were applied and the following measurements of both breasts were estimated: volumes (vol), nipple–sternal notch distance (A-sn), difference between nipples levels (A-A’), nipple–midline distance (A-ml), difference between inframammary fold levels (IF-IF’), distance between inframammary fold and nipple (IF-A), distance between inframammary fold apex and midline (IF-ml). Additionally, proportions of the breasts’ quadrants were evaluated. For vertical parameters, we used the ratio of distance from the upper pole apex to the nipple and distance from the inframammary fold apex to the nipple (UP:LP). For horizontal parameters, we used the ratio of distance from the breast’s lateral border to the nipple and distance from the breast’s medial border to the nipple (LB:MB). As an aesthetic standard, the following proportions were accepted: for UP:LP—45:55 and for LB:MB—40:60 [24,25]. All measurements were performed preoperatively and 6 months after the surgery by one surgeon from the operative team (T.Z.), twice in an interval of several days. The average value of two measurements was accepted as a result. Asymmetries of all variables were calculated (asy-vol, A-A’, asyA-sn, asyA-ml, IF-IF’, asyIF-A, asyIF-ml). The study protocol was approved by the Bioethics Committee of the Medical University of Łódź (RNN/330/19/KE).

### 2.2. Statistical Analysis

To compare the measurements before and after the surgery, average proportions before and after surgery, and proportions before the surgery with the aesthetic reference values, we used the t-test for related samples. When the distribution of asymmetries before and after the surgery was normal, t test was also applied for their comparisons, otherwise the Wilcoxon test was used. To assess the correlation between the postoperative difference in the position of the nipple (A-A’) and postoperative differences in breast volumes (asy-vol) and clinical and metric breasts’ variables Spearman’s correlation analysis was used. To compare variables of patients with postoperative volume asymmetry greater than the postoperative mean with women with lower postoperative asymmetry, we used t-test (normal distribution) and the Mann–Whitney test (other cases). In all measurements, the normality of distribution of each variable was tested using the Shapiro–Wilk test and a *p* < 0.05 was accepted as a level of significance. Analyses were performed using the Statistica package (v13, StatSoft, Cracow, Poland)

## 3. Results

### 3.1. Participants’ Characteristics

The average age of analyzed women was 37.3 years (SD = 10 years). The average BMI was 25 kg/m^2^ (SD = 2 kg/m^2^). All women were qualified for breast reduction due to health conditions such as back pain and dermatologic conditions; all had endocrine disturbances excluded. The average weight of the excised tissues was 685 g (SD = 272 g) for right breast and 686 g (SD = 277 g) for left breast.

The values of preoperative and postoperative asymmetries of the analyzed breast measurements and volumes are presented in Table 1. All asymmetries, but for asyIF-ml, decreased significantly after breast reduction. Additionally, we also observed improvement in postoperative upper pole to lower pole proportions (UP:LP), but statistically we did not achieve the desired proportions proposed by Malucci P et al. [17,18]. Postoperative proportions of medial and lateral portions (LB:MB) of the breast differed significantly from the desired proportions proposed by Lewin et al. [19] (Table 2) and the surgery did not even bring it closer to the suggested values.

### 3.2. Predictors of Postoperative Nipples Location Asymmetry

Postoperative asymmetry of nipple level was associated with preoperative asymmetry of IF-ml; however, logistic regression did not detect any preoperative measurement influencing postoperative nipple position asymmetry (Table 3).

### 3.3. Predictors of Postoperative Volume Asymmetry 

Postoperative asymmetry of breast volume was not associated with any of the analyzed clinical and metric preoperative variables (Table 4). To determine the differences in the analyzed variables in women who presented higher postoperative breast asymmetry (higher than the average—52 cc in our group) (*n* = 29), we compared them with women who presented postoperative volumetric asymmetry lower than the average. We found that higher preoperative asyIF-ml increased the risk of postoperative volume asymmetry that is above the average (52 cc) (OR = 2.04). (Table 5).

## 4. Discussion

In any type of breast surgery, postoperative symmetry is one of the most desirable outcomes for both patient and the surgeon. In this study, we aimed to verify if preoperative breast asymmetry is a predictor of postoperative asymmetry in women undergoing breast reduction. We focused on analyzing clinical variables’ influence and preoperative breast measurement asymmetries’ effect on the postoperative asymmetry of breast volume and nipple positions. For the assessment an objective, metric methods were applied. We did not find any clinical characteristic to be associated with postoperative breast asymmetry and, among breast measurements, only asymmetry of inframammary fold distance to the midline appeared to influence postoperative volumetric asymmetry.

Numerous studies have shown that reduction mammoplasty is effective in relieving the clinical symptoms of macromastia and improving the quality of life of patients [26,27,28,29]. Furthermore, many studies found the influence of various variables such as BMI > 30 kg/m^2^, age > 50 years, weight of the resected tissues, distance of NAC transposition, and smoking on the incidence of postoperative complications [26,27,28,29], but there are limited studies analyzing the influence of these factors on postoperative breast asymmetry. In this study, none of the analyzed factors turned out to be significant in the context of postoperative asymmetry of numerous metric features and breast volume. However, we were not able to fully verify BMI as a possible factor increasing the risk of postoperative breast asymmetry as all our participants had BMI less than 30 kg/m^2^, as this is our criterion for a patient’s qualification for breast reduction surgery.

Hudson et al. (2019) analyzed anthropometrically breasts of women with macromastia and emphasized the correlation between increased BMI values and a decrease in IMF levels and lateral nipple displacement from the breast meridian. Our results add to the knowledge that this displacement is often asymmetric; however, surgery significantly improves the symmetry and location of nipples. The authors also advised parenchymal sutures to reinforce the IMF to improve long-term aesthetic results preventing pseudoptosis in patients undergoing breast reduction using Wise Pattern and superomedial pedicles. Using this technique, they found lengthening of the nipple to IMF distance in only three patients. Using maneuvers on IMF may result from a characteristic of the group the authors analyzed, as the average BMI of their participants was 31 kg/m^2^ (14 of 25 patients included in this study had a preoperative BMI over 30 kg/m^2^) [31,32]. We did not observe statistically significant difference between preoperative and postoperative asymmetry of IMF apex distance to the midline. This indicates that our maneuvers on this structure are not common, probably due to the specificity of our studied sample, including only normal-weighted women. BMI range in our study group was rather small, so the validity of analyzing this feature in our models was limited. Our finding concerning higher postoperative volumetric asymmetry in women with higher preoperative asymmetry of the apex of IMF to midline can be related to the asymmetric breast footprint indicated by this feature. Asymmetric breast “basis” may cause problems with the clinical assessment of volumetric symmetry intraoperatively.

For volume measurements, Yang et al. (2021) proposed an objective method based on a handheld 3D scanner to assist intraoperative symmetry assessment, aiming to achieve better aesthetic results in reduction mammaplasty. In the study group, the surgeon performed surgery using the technique of vertical scar reduction mammaplasty with a superior pedicle and if the volumetric difference of the breasts was more than 50 cm^3^, an appropriate volume of the larger breast was further removed according to the measurement results. In the control group, intraoperative breast symmetry was dependent on the surgeon’s visual estimations. The breast volume difference in the study group was significantly smaller than in the control group (39.1 vs. 113.3 cc, *p* = 0.001). Similar difference was found for asymmetry of the nipple to inframammary fold distance (2.79 vs. 7.43 mm, *p* = 0.01) [33]. As in our results, the main factor of postoperative asymmetry appeared to be the “sensitivity” of the surgeon’s eye or availability of tools for such intraoperative assessment, not any of the clinical or metric preoperative features.

Henseler et al. (2011, 2012, 2015) analyzed accuracy of breast volume prediction using water displacement techniques comparing 3D scanning using multiple stereo cameras. Difference in results did not exceed 40 cc [34,35,36]. This difference was less than the volume difference detectable by the human eye, which was quoted as 50 cc by subjective judgment reported by Sigurdson et al. [37]. In our patients, the average value of preoperative volume asymmetry was 90 cc, which indicated a visible breast asymmetry, while after breast reduction the mean difference was 52 cc. According to the literature, such asymmetry seems to be beyond the human eye, so breasts can be regarded as clinically symmetric.

Swanson et al. (2011) analyzed 82 international publications on mastopexies and breast reductions. Breast and upper pole projection were not significantly enhanced by any of the mastopexy or reduction mammaplasty procedures or using fascial sutures or auto-augmentation techniques. There was no significant difference in outcomes compared with the follow-up time, amount of resection, year of publication, or geographic region. Authors concluded that existing mastopexy/reduction techniques do not significantly increase breast or upper pole projection, and fascial sutures and auto-augmentation techniques are also ineffective [38]. Our results showed that, after surgery, breast upper and lower pole proportions indicate a reduction of the upper pole, which may result from a significant NAC transfer. Breast reduction significantly improves breast proportions clinically, but the “aesthetic norms” proposed by Mallucci et al. [17,18] have limited application for evaluating breasts after reduction mammaplasty procedures.

The study has some limitations. We included only Caucasian women; thus, the results may be ethnically specific and may not allow for generalization. Our sample included only normal-weighted women, and obesity that could increase the risk of postoperative complications and affect the results was an exclusion criterion. Measurements were performed by a researcher from the operative team, which could also be regarded as a limitation. Furthermore, it would be worthwhile to compare the long-term evaluation of postoperative effects and their evolution, e.g., after 2 years post breast reduction.

## 5. Conclusions

Postoperative breast asymmetry after breast reduction is not related to preoperative asymmetries nor clinical variables; however, asymmetry of the inframammary fold apex to the midline may be the factor affecting postoperative volumetric asymmetry. It may be suggested that surgical techniques including preoperative planning and intraoperative symmetry assessment are the main predictors of postoperative symmetry. To verify our findings, more research concerning this aspect is needed in larger and more clinically diverse groups.

## Figures and Tables

**Figure 1 ijerph-20-03780-f001:**
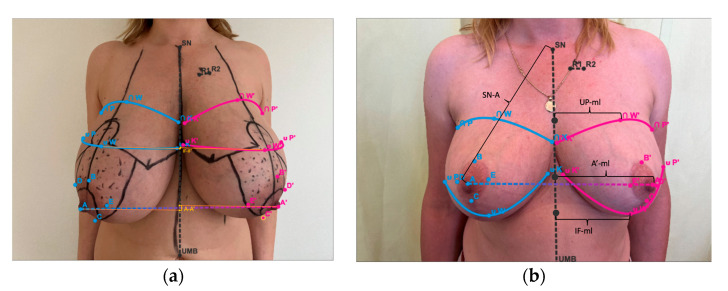
Breast Idea Quick Assessment: (**a**) preoperative measurements; (**b**) postoperative measurements (6 months after the surgery). Note. For better clarity of figures, differences of levels are shown only on (**a**) and measurements only on (**b**), but all variables were estimated on pre and postoperative photographs: A-A’—difference between nipple levels; IF-IF’—difference between inframammary folds levels; SN-A—nipple to sternal notch distance; IF-ml—inframammary fold apex to midline distance; UP-ml—upper pole apex to midline distance.

**Figure 2 ijerph-20-03780-f002:**
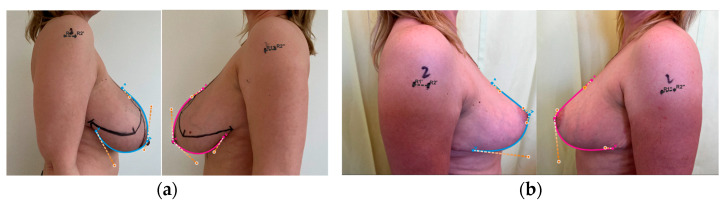
Breast Volume Estimator: (**a**) preoperative volume estimation; (**b**) postoperative volume estimation (6 months after the surgery).

**Table 1 ijerph-20-03780-t001:** Comparison of preoperative and postoperative (6 months after breast reduction) asymmetries.

	Preoperative			Postoperative			
	Mean	±	SD	Mean	±	SD	*p*
A-A’	1.46	±	1.05	0.74	±	0.54	<0.0001
IF-IF’	0.63	±	0.45	0.45	±	0.34	0.0154
asyA-ml	1.11	±	0.80	0.74	±	0.66	0.0048
asyA-sn	1.46	±	1.48	0.58	±	0.40	<0.0001
asyIF-A	1.32	±	1.22	0.49	±	0.40	<0.0001
asyIF-ml	0.91	±	0.75	0.82	±	0.71	0.4094
asy-vol	90.32	±	82.95	51.92	±	3.95	0.0006

*t*-test.

**Table 2 ijerph-20-03780-t002:** Comparison of preoperative and postoperative (6 months after breast reduction) upper pole to lower pole proportions and medial to lateral portions of breasts.

	Preoperative	Postoperative	Reference Values	Preoperative vs. Postoperative	Postoperative vs. Reference Values
				t;p	t;p
Mean UP:LP	66:34	56:44	45:55	9.04; <0.0001	12.25; <0.0001
Mean LB:MB	35:65	23:77	40:60	11.59; <0.0001	−18.41; <0.0001

*t*-test.

**Table 3 ijerph-20-03780-t003:** Relationship between the postoperative asymmetry of nipple position (6 months after breast reduction) and clinical and preoperative metric variables.

Postoperative Asymmetry A-A’	R	t	p
vs.			
Age	0.026	0.22	0.8299
Height	−0.021	−0.17	0.8620
Weight	0.044	0.36	0.7167
BMI	0.015	0.13	0.8990
Mean tissue resection	0.195	1.65	0.1030
A-A’	0.075	0.63	0.5321
IF-IF’	−0.066	−0.55	0.5840
asyA-ml	0.073	0.61	0.5454
asyA-sn	0.204	1.73	0.0876
asyIF-A	0.099	0.83	0.4079
asyIF-ml	0.326	2.86	0.0056
asy-vol	0.057	0.47	0.6375

*t*-test, *R-Spearman’s correlation analysis*.

**Table 4 ijerph-20-03780-t004:** Relationship between the postoperative volume asymmetry (6 months after breast reduction) and clinical and preoperative metric variables.

Postoperative Volume Asymmetry	R	t	*p*
vs.			
Age	−0.002	−0.02	0.9859
Height	0.121	1.01	0.3163
Weight	−0.004	−0.03	0.9762
BMI	−0.074	−0.62	0.5388
Mean tissue resection	0.147	1.24	0.2199
A-A’	0.057	0.48	0.6347
IF-IF’	0.048	0.40	0.6907
asyA-ml	0.115	0.96	0.3385
asyA-sn	0.027	0.23	0.8212
asyIF-A	0.060	0.50	0.6204
asyIF-ml	0.183	1.55	0.1262
asy-vol	0.166	1.40	0.1667

*t*-test, R-Spearman’s correlation analysis.

**Table 5 ijerph-20-03780-t005:** Comparison of clinical and metric variables in women with postoperative (6 months after breast reduction) volume asymmetry higher than the average (52 cc) versus lower than the average.

	Postoperative Volume Asymmetry (Higher than the Average)						
		N	Mean	±	SD	Z/t	p
Age		29	37.55	±	10.91	0.1679	0.8671
Height		29	1.67	±	0.06	0.3641	0.7169
Weight		29	67.83	±	6.65	−0.2331	0.8164
BMI		29	24.31	±	1.57	−0.6346	0.5278
Mean tissue resection		29	714.78	±	275.87	0.7519	0.4547
Preoperative A-A’	*	29	1.60	±	1.30	0.3217	0.7477
Preoperative IF-IF’	*	29	0.69	±	0.44	1.0646	0.2871
Preoperative asyA-ml	*	29	1.29	±	0.86	1.5852	0.1129
Preoperative asyA-sn	*	29	1.41	±	1.24	−0.1287	0.8976
Preoperative asyIF-A	*	29	1.29	±	1.24	−0.4797	0.6314
Preoperative asyIF-ml	*	29	1.13	±	0.73	2.5620	**0.0104**
Preoperative asy-vol	*	29	102.06	±	101.63	0.4621	0.6440

*t*-test, *** Mann-Whitney test, bold font—statistical significance.

## Data Availability

Data available on request from the corresponding author.

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
