# Peer review of "Is Preoperative Asymmetry a Predictor of Postoperative Asymmetry in Patients Undergoing Breast Reduction?"

_ijerph, 2023, doi:10.3390/ijerph20053780_

Round 1
Reviewer 1 Report
I read with great interest the paper "Is preoperative asymmetry a predictor of postoperative asymmetry in patients undergoing breast reduction?".
I have few suggestions:
- Page 2, line 85, please move "Participants’ characteristics" to the results.
- The authors should add details regarding the surgical technique; do they perform septum-based breast reduction?
- The authors should consider adding and discussing these citations in the manuscript (PMID: 32128706; PMID: 19182600).
Author Response
Dear Editor and Reviewers,
Thank you for your interest in our manuscript entitled " Is Preoperative Asymmetry A Predictor of Postoperative Asymmetry in Patients Undergoing Breast Reduction?”. We would like to thank anonymous Reviewers for their valuable comments and reviews, which helped to improve the manuscript. In this revision we addressed all your comments (in the text changes are marked in red). Additionally, according to Editor’s suggestion we added additional paragraph to increase word count to meet the Journal criteria. We hope that this revision meets with your approval.
Sincerely,
The Authors
Responses to Reviewer’s comments:
Reviewer 1
I read with great interest the paper "Is preoperative asymmetry a predictor of postoperative asymmetry in patients undergoing breast reduction?".
I have few suggestions:
- Page 2, line 85, please move "Participants’ characteristics" to the results.
Done.
- The authors should add details regarding the surgical technique; do they perform septum-based breast reduction?
In the description of our surgical technique we specified the type of our pedicle and resection (“a patient qualified for the Wise pattern reduction mammaplasty with superomedial dermal pedicle (random vascularization) for the transposition of the NAC with the gland resection caudal to the Wuringer’s septum in the inferior part of the breast followed by a triangular resection lateral to the pedicle”).
- The authors should consider adding and discussing these citations in the manuscript (PMID: 32128706; PMID: 19182600).
The suggested references were included in the introduction in the description of different surgical techniques.
Reviewer 2 Report
Please make more clear in tables and text of material and methods that “postoperative “ signifies
6 months post op
Author Response
Dear Editor and Reviewers,
Thank you for your interest in our manuscript entitled " Is Preoperative Asymmetry A Predictor of Postoperative Asymmetry in Patients Undergoing Breast Reduction?”. We would like to thank anonymous Reviewers for their valuable comments and reviews, which helped to improve the manuscript. In this revision we addressed all your comments (in the text changes are marked in red). Additionally, according to Editor’s suggestion we added additional paragraph to increase word count to meet the Journal criteria. We hope that this revision meets with your approval.
Sincerely,
The Authors
Responses to Reviewer’s comments:
Reviewer 2
Please make more clear in tables and text of material and methods that “postoperative “ signifies 6 months post op
This was specified.
Reviewer 3 Report
With interest I've read this study concerning the instance if preoperative breast asymmetry is a predictor of postoperative asymmetry in women undergoing breast reduction. I find the article well written and interesting for the general audience. I only suggest to include a picture where the measurements estimated are more clearly illustrated.
Author Response
Dear Editor and Reviewers,
Thank you for your interest in our manuscript entitled " Is Preoperative Asymmetry A Predictor of Postoperative Asymmetry in Patients Undergoing Breast Reduction?”. We would like to thank anonymous Reviewers for their valuable comments and reviews, which helped to improve the manuscript. In this revision we addressed all your comments (in the text changes are marked in red). Additionally, according to Editor’s suggestion we added additional paragraph to increase word count to meet the Journal criteria. We hope that this revision meets with your approval.
Sincerely,
The Authors
Responses to Reviewer’s comments:
Reviewer 3
With interest I've read this study concerning the instance if preoperative breast asymmetry is a predictor of postoperative asymmetry in women undergoing breast reduction. I find the article well written and interesting for the general audience. I only suggest to include a picture where the measurements estimated are more clearly illustrated.
The measurements were presented more clearly on the photographs /marked with brackets/.